# Age-Adjusted D-Dimer Levels May Improve Diagnostic Assessment for Pulmonary Embolism in COVID-19 Patients

**DOI:** 10.3390/jcm11123298

**Published:** 2022-06-09

**Authors:** Michał Machowski, Anna Polańska, Magdalena Gałecka-Nowak, Aleksandra Mamzer, Marta Skowrońska, Katarzyna Perzanowska-Brzeszkiewicz, Barbara Zając, Aisha Ou-Pokrzewińska, Piotr Pruszczyk, Jarosław D. Kasprzak

**Affiliations:** 1Department of Internal Medicine & Cardiology, Medical University of Warsaw, Lindleya 4 St., 02-005 Warsaw, Poland; michal.machowski@wum.edu.pl (M.M.); magdalena.galecka-nowak@wum.edu.pl (M.G.-N.); krzyskasia@op.pl (K.P.-B.); aishaou@gmail.com (A.O.-P.); piotr.pruszczyk@wum.edu.pl (P.P.); 2I Department of Cardiology, Bieganski Hospital, Medical University of Lodz, Kniaziewicza 1/5 St., 91-347 Lodz, Poland; anna.polanska@onet.pl (A.P.); a.mdachnowska@gmail.com (A.M.); dmbarbarazajac@gmail.com (B.Z.); kasprzak@ptkardio.pl (J.D.K.)

**Keywords:** acute pulmonary embolism, venous thromboembolic disease, COVID-19, COVID-19-associated coagulopathy

## Abstract

Introduction: SARS-CoV-2 infection leads to a hypercoagulable state. The prevalence of pulmonary embolism (PE) seems to be higher in this subgroup of patients. Patients and methods: We combined data from two tertiary referral centers specialized in the management of PE. The aims of this study were as follows: (1) to evaluate the prevalence of PE among a large population of consecutive patients admitted for COVID-19 pneumonia in two centers, (2) to identify a plasma D-dimer threshold that may be useful in PE diagnostic assessment, (3) to characterize the abnormalities associated with PE and mortality in COVID-19 patients. Results: The incidence of symptomatic acute PE was 19.3%. For diagnosing PE in COVID-19 patients, based on ROC curve analysis, we identified a D-dimer concentration/patient’s age ratio of 70, which improved D-dimer diagnostic capacity for PE and led to a reclassification improvement of 14% (NRI 0.14, *p* = 0.03) when compared to a cut-off level of 1000 ng/mL. Especially in severe COVID-19 lung involvement, D-dimer/age ratio cut-off equal to 70 was characterized by high diagnostic feasibility (sensitivity, specificity, negative predictive value, positive predictive value of 83%, 94%, 96%, and 73%, respectively). Apart from PE status, lung involvement and troponin T concentration were also independent predictors of in-hospital mortality. In the subgroup of PE patients, mortality was comparable with non-PE patients (19/88 (21.5%) vs. 101/368 (27.4%) for non-PE, *p* = 0.26) and was associated with older age, higher Bova scores, and higher troponin T concentrations. Age was the sole independent predictor for mortality in this subgroup. Conclusions: PE in COVID-19 patients is common, but it may not influence mortality when managed at a specialized center. In suspected PE, age-adjusted D-dimer levels (upper limit of normal obtained from the formula patient’s age × 70) may still be a useful tool to start the diagnostic workup. In COVID-19 patients without PE, older age, more extensive parenchymal involvement, or higher D-dimer levels are factors predicting mortality.

## 1. Introduction

Severe cases of viral pneumonia, such as severe acute respiratory syndrome coronavirus 2 (SARS-CoV-2) infection predispose patients to a hypercoagulable state, which clinically manifests as an observed increase in the overall rate of venous thromboembolism, including in-situ pulmonary thrombosis [1,2,3]. Among alternations in coagulation parameters caused by COVID-19-associated coagulopathy, increased levels of fibrin degeneration products—namely, D-dimer—are of special interest to this study [3]. 

The standard PE diagnostic protocol includes D-dimer concentration testing and imaging studies [4]. PE, in the setting of acute viral pneumonia, has not been studied before, and, therefore, the standard thresholds for diagnostic markers cannot be extrapolated onto the COVID-19 population [3,5,6]. There is an on-going discussion on the optimal D-dimer cut-off levels for PE diagnosis in COVID-19 patients.

The aims of this study were as follows: (1) to evaluate the prevalence of PE among a large population of consecutive patients admitted for COVID-19 pneumonia, (2) to identify a plasma D-dimer threshold that may be useful in PE diagnostic assessment, and (3) to characterize the abnormalities associated with PE and mortality in COVID-19 patients.

## 2. Methods

This is a retrospective analysis of consecutive patients hospitalized, from October 2020 to May 2021, with radiologically confirmed COVID-19 pneumonia and oxygen desaturation <94%. We combined data from two tertiary referral cardiac units specialized in the management of PE. Patients were analyzed depending on PE status and followed-up for the occurrence of PE and for mortality.

The study was approved by local institutional ethics committees. 

### 2.1. Laboratory Analysis

COVID-19 was diagnosed when acute respiratory symptoms or an exacerbation of chronic respiratory symptoms were present and one of the following occurred: SARS-CoV-2 target genes were detected using a reverse-transcriptase polymerase chain reaction (RT-PCR) assay (CovGenX) from biological material, collected using nasopharyngeal swabs or with a positive immunochromatographic lateral flow test, detecting the target nucleocapsid protein of SARS-CoV-2 from nasopharyngeal swabs (Abbott, Chicago, IL, USA).

Plasma concentrations of troponin T were measured as part of a standard diagnostic protocol using a high-sensitivity automated sandwich electrochemiluminescence immunoassay (Roche Diagnostics GmbH, Mannheim, Germany). Values above 0.014 ng/mL were considered elevated.

D-dimer concentrations were quantitatively measured, on the day of admission, as part of a standard diagnostic protocol using an automated enzyme-linked fluorescent assay (VIDAS D-dimer Exclusion, bioMerieux, Marcy-l’Étoile France) or using a turbidimetric immunoassay (HemosIL, Werfen, Spain); depending on patient location, both had a reference range of values up to 500 ng/mL.

### 2.2. Imaging Studies

All patients underwent chest computed tomography (CT) scans for the diagnosis and evaluation of the severity of pulmonary lesions caused by SARS-CoV-2. In the case of a clinical suspicion of PE, a contrast-enhanced multi-slice computed tomography of pulmonary arteries (CTPA) was performed. PE was confirmed when thromboemboli were visualized, at least, at the level of segmental pulmonary arteries. CTPA was performed using an 80-row Toshiba Aquilion Prime CT scanner (Toshiba Medical Systems, Otawara, Japan) or an Aquilion One Genesis scanner (Canon Medical Systems, Otawara, Japan). The results of CT studies were adjudicated by two radiology specialists.

Transthoracic echocardiography was performed within 24 h after PE diagnosis. The examination was performed according to the guidelines of the American Society of Echocardiography and the European Association of Cardiovascular Imaging, with modifications according to the pandemic recommendations of the Polish Cardiac Society [7,8,9]. All examinations were performed by a physician certified in echocardiography using the Philips iE33 (Philips Medical Systems, Andover, MA, USA) or EPIQ 7 system (Phillips, Eindhoven, NL, USA) or Vivid e95 (GE Healthcare, IL, USA). Right ventricle (RV) pressure overload was defined as the presence of any of the following: tricuspid regurgitation peak pressure gradient > 30 mmHg, right ventricle/left ventricle ratio in the apical four-chamber view > 1.0, or the presence of McConnell’s sign. 

### 2.3. Bova Score Calculation

The Bova score is a four-variable tool used to stratify normotensive patients with confirmed PE to identify patients at intermediate and high risk for complications and mortality associated with PE. The variables include: heart rate > 110/min, systolic blood pressure <100 mmHg, elevated cardiac troponin levels, and signs of RV dysfunction in imaging studies [10]. Bova score values were calculated for each patient using data obtained from the database.

### 2.4. Data Storing

A dedicated database for storing patient data was used in both centers. We collected data of COVID-19 patients, at least 18 years of age, who underwent a chest CT at admission and required hospitalization. Excluded are patients who did not consent to participating in the study. Information was collected on demographic and medical characteristics, including age, sex, body mass index (BMI), length of hospitalization, anticoagulant treatment, and extent of pulmonary lesions in CT (severe ≥ 50%, non-severe < 50%). Noted biochemical parameters included: D-dimer and high-sensitivity troponin T concentrations. In the subgroup of PE patients, the presence of right ventricle dysfunction, total Bova score, and need for thrombolytic therapy were recorded.

Hospitalized COVID-19 patients received pharmacological thromboprophylaxis. PE was treated following the European Society of Cardiology (ESC) guidelines [4]. 

### 2.5. Statistical Analysis

Data are expressed as parameter or median, followed by interquartile range. The Kolmogorov–Smirnov test was used to identify continuous variables with a skewed distribution which were then compared using the Mann–Whitney U test. Categorical data were compared using Chi2 or Fisher’s exact test, depending on the sample size. For all performed tests, *p*-values of <0.05 were considered significant. All tests were two-tailed. Receiver operating characteristic (ROC) analysis was used to determine the area under the curve (AUC) for D-dimer concentrations in diagnosing the occurrence of PE. To validate whether the proposed threshold could improve the diagnostic capacity of D-dimer, in this population, the net reclassification improvement was calculated using the method described by Pencina et al. [11].

To explore the risk factors associated with in-hospital death, multivariable logistic regression models were used. Considering the total number of deaths in our study, and to avoid overfitting in the model, four variables were chosen for multivariable analysis, based on previous findings and clinical constraints. 

Analyses were performed using the STATISTICA 13 data analysis software system (TIBCO Software Inc., Palo Alto CA, USA) and the MedCalc software system (MedCalc Software Ltd., Ostend, Belgium).

## 3. Results

456 patients with COVID-19 were included in the study (256 patients from Warsaw and 200 from Łódź), of which 88 (19.3%) had confirmed symptomatic pulmonary embolism. None of the patients from the original non-PE group developed PE during in-hospital observation. The median follow-up time was 10 days (25th–75th percentile 5–14.5). The overall mortality was high at 120/456 pts (26.3%). We found no major difference in clinical characteristics, PE frequency, and clinical outcome between participating centers. Relevant patient characteristics are presented in Table 1.

Independently of PE status, patients with a complicated clinical course were older, were characterized by higher troponin T and D-dimer concentrations, and had more parenchymal involvement in CT scans. Moreover, age, lung involvement, and troponin T concentration were all independent predictors of mortality in the whole population.

In the subgroup of PE patients, mortality was comparable with non-PE patients (19/88 (21.5%) vs. 101/368 (27.4%) for non-PE, *p* = 0.26). PE patients were characterized by higher D-dimer concentrations, higher D-dimer/patient’s age ratios, with no differences in age alone or troponin T levels when compared with non-PE patients. Within the subgroup of PE patients, mortality was associated with older age, higher Bova scores, and higher troponin T concentrations. Age was the sole independent predictor for mortality in this subgroup. Full data is available in Table 1, Table 2 and Table 3. The flow of patients is presented in Figure 1.

All PE patients received, at least, anticoagulation, seven patients received systemic thrombolysis. The decision to start aggressive therapy was left to the managing team based on clinical evaluation and existing ESC guidelines. Only one COVID-19 PE (−) patient was left without pharmacological venous thromboembolism (VTE) prophylaxis.

For diagnosing PE in COVID-19 patients, based on ROC curve analysis, we identified a D-dimer concentration/age ratio of 70 as the threshold for continuing further testing for PE (negative predictive value (NPV) of 90%, positive predictive value (PPV) of 60%). The ROC curve is presented in Figure 2. The net reclassification improvement for the proposed threshold vs. 1000 ng/mL was 0.14, standard error 0.7, *p* = 0.03.

Finally, we looked at the usefulness of D-dimer concentrations in suspecting PE depending on lung involvement in a standard chest CT. Parenchymal involvement ≥ 50% was considered severe, values below 50% were considered non-severe. In patients with severe lung tissue involvement, the proposed D-dimer/age ratio cut-off equal to 70 was characterized by high diagnostic feasibility (sensitivity, specificity, NPV, PPV) of 83%, 94%, 96%, and 73%, respectively). In non-severe parenchymal involvement, the afore mentioned threshold was characterized by a sensitivity of 45%, specificity of 89%, NPV of 87%, and PPV of 49%. For higher sensitivity, while maintaining satisfactory specificity, a lower cut-off value characterized by the highest Youden’s index was chosen—D-dimer/age > 20 (sensitivity 75%, specificity 65%, NPV 92%, PPV 35%). ROC curves are presented in Figure 3. 

## 4. Discussion

The key findings emerging from our retrospective analysis concern the utility of D-dimer in diagnosing PE in the setting of acute viral pneumonia caused by SARS-CoV-2. The obvious barriers connected with further diagnostic testing for PE in infectious patients warrant searching for more feasible diagnostic protocols. Previously, D-dimer level measurement was recommended to exclude PE in patients with non-high clinical probability of acute PE. However, in COVID-19 patients, markedly elevated D-dimers were reported to indicate patients with a high probability of VTE and, therefore, increased D-dimer levels, which may be proposed to initiate the diagnostic workup for VTE [2]. There is an ongoing discussion on the optimal cut-off value that should mandate VTE diagnostic assessment, with some authors suggesting values previously applied for PE and non-adjusted higher than conventional thresholds, such as 1000 ng/mL [6]. Since the most commonly used conventional non-age-adjusted threshold for D-dimer levels (i.e., upper limit of normal 500 ng/mL) is not adequate in this patient population, and the notion of employing D-dimer testing in PE diagnostic assessment in COVID-19 patients has even been discouraged [12], we examined other, non-standard thresholds—namely, the cut-offs calculated from the formula: D-dimer level/patient’s age, with the threshold > 70. In our study, employing the modified, higher threshold led to a reclassification improvement of 14% (NRI 0.14, *p* = 0.03) when compared to the suggested, in literature, cut-off equal to 1000 ng/mL. 

In the recently published Co-LEAD study, authors stratified D-dimer levels by the extent of parenchymal lung involvement, concluding two different D-dimer thresholds for continuing to CTPA [13]. In this study, we sought a similar approach and found that a dichotomous threshold, depending on lung tissue involvement (D-dimer/age > 20 for non-severe lesions, D-dimer/age > 70 for severe lesions), may be diagnostically useful. 

Our second major observation is that, in SARS-CoV-2 pneumonia patients, despite thrombosis prophylaxis, the co-existence of symptomatic PE is common, but it does not lead to differences in mortality between PE and non-PE patients, and thus, it may not influence the survival. It should be underlined that both centers contributing to the current study are referral cardiovascular centers and are experienced in VTE management. Interestingly, data on the effect of venous thromboembolism on mortality in COVID-19 patients remain ambiguous. However, a similar observation regarding survival was made in a meta-analysis based on over 5000 patients and in two smaller studies [14,15,16]. Age, D-dimer levels, and the extent of parenchymal lung involvement all predicted mortality in the COVID-19 (+) PE (−) subgroup of patients. The role of elevated D-dimer concentrations, at admission in prognosticating in-hospital mortality in COVID-19, was extensively reported, with thresholds varying between 1000 ng/mL up to even 4800 ng/mL [17,18,19,20,21]. This study also confirms that observation. In PE (+) patients, D-dimer levels were markedly higher than in non-PE patients; however, the only independent mortality predictor was age. 

Another noteworthy issue is the usefulness of guiding anticoagulation treatment with adjusted D-dimer levels [22], especially in the extended treatment (non-acute) phase. It has been reported that elevated D-dimer levels may be positively correlated with bleeding risk in non-COVID-19 PE patients [23], displaying potential in both prognosticating mortality and hemorrhagic complications during PE treatment.

Finally, it may be plausible that, in this group of patients, the extent of SARS-CoV-2 infection (myocardial damage) was the main driver of elevated troponin T levels, rather than PE status. Hence, troponin may not be a reliable indicator of right heart strain in COVID positive PE patients. However, taking into account certain limitations of this study, namely that consecutive patients, regardless of the risk of early PE mortality defined according to the ESC criteria, were included, we abstain from drawing any definitive conclusions based on the aforementioned finding.

Several study limitations should be acknowledged. Firstly, this is a relatively small, two-center study conducted in tertiary cardiac centers specialized and experienced in PE management. This may lead to selection bias, in terms of the mortality and the characteristics of enrolled patients, and may not be fully representative of a broader population. Importantly, in the course of COVID-19 thrombi detectable in pulmonary arteries may represent both true embolism and in situ thrombosis, and for this study, we did not attempt differentiating these subtypes. Moreover, consecutive patients, regardless of the risk of early PE mortality, defined according to the ESC criteria, were included. It should be underlined that the study end-point was all-cause death. All the more so, we must robustly underline the exploratory nature of this study. 

## 5. Conclusions

Our study demonstrates a high prevalence of PE and COVID-19 in the Polish patient population hospitalized due to SARS-CoV-2 pneumonia. However, PE does not influence in-hospital survival when treated in a referral center. In our opinion, age adjusted D-dimers levels (upper limit of normal obtained from the formula patient’s age × 70) may be used to start diagnostic workup for PE.

## Figures and Tables

**Figure 1 jcm-11-03298-f001:**
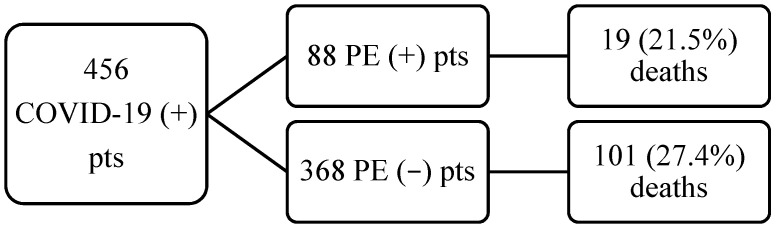
Flow of patients in the study.

**Figure 2 jcm-11-03298-f002:**
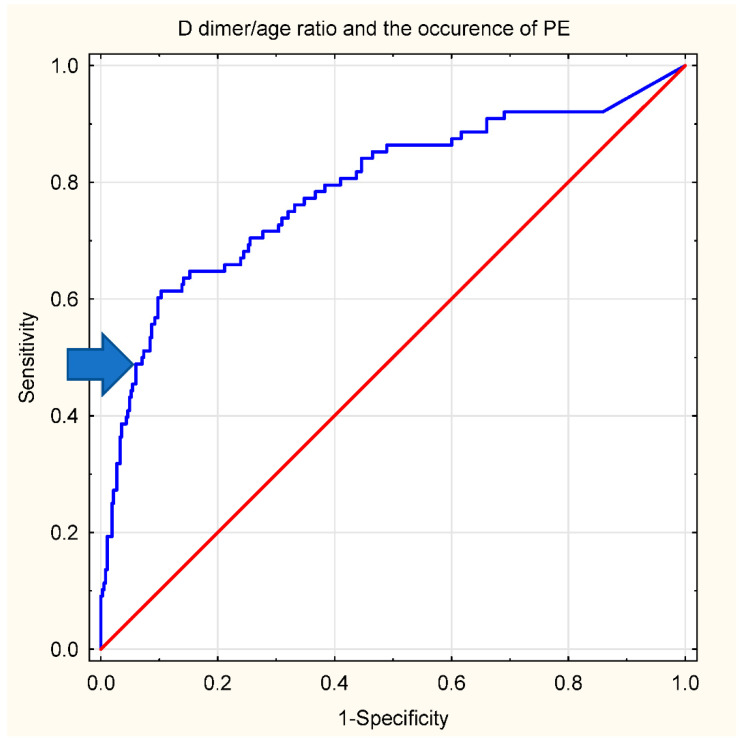
ROC curve for D-dimer/age ratio and occurrence of PE. AUC 0.788 (95% CI 0.727–0.848). The proposed cut-off is marked with an arrow (D-dimer/age = 70). Pts—patients. The red line is the no-effect line.

**Figure 3 jcm-11-03298-f003:**
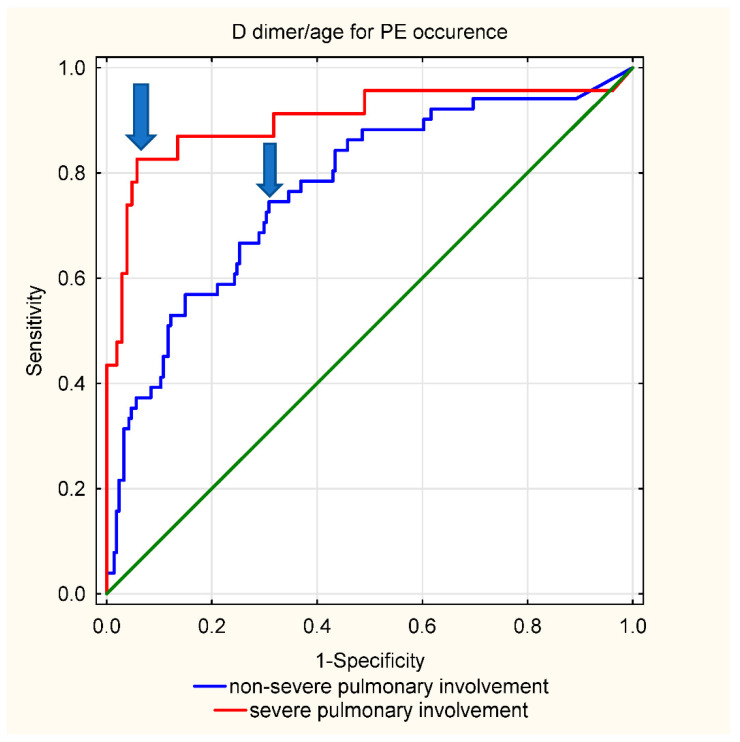
ROC curve for D-dimer/age ratio and occurrence of PE depending on the extent of pulmonary lesions, i.e., ≥50% or <50%. AUC for non-severe involvement = 0.769 (95% CI 0.694–0.845); AUC for severe involvement = 0.902 (95% CI 0.810–0.994), *p* = 0.029 for difference in AUC. The proposed thresholds are marked with arrows (D-dimer/age = 70, proposed for severe involvement, or D-dimer/age = 20 for non-severe involvement). The green line is the no-effect line.

**Table 1 jcm-11-03298-t001:** Characteristic of the study group and a comparison of COVID-19 patients with pulmonary embolism (PE+) and without pulmonary embolism (PE−). Data are presented as number (median, percentage) followed by interquartile range, where applicable.

	All N = 456	PE (−) N = 368	PE (+) N = 88	*p* Value PE (+) vs. PE (−)
Age (years)	69 (61–78)	69 (61–77)	69 (58–78.5)	0.98
Female (n, %)	162, 35.5	138, 37.5	24, 27.3	0.7
Length of hospitalization (days)	10 (5–14.5)	9 (4–14)	11 (6–15.5)	0.01
CT lung involvement (%)	40 (20–65)	40 (20–60)	47.5 (20–60)	0.87
D-dimer (ng/mL)	1317 (728–3948)	1117.5 (625.5–2120)	6764 (1973–21,548)	<0.001
D-dimer/age ratio	16.9 (7.8–42.6)	13.8 (6.8–28.7)	87 (21.9–303.0)	<0.001
Troponin T (ng/mL)	0.027 (0.012–0.087)	0.027 (0.012–0.086)	0.027 (0.012–0.09)	0.66
Mortality (n;%)	120; 26	101; 27.4	19; 21.5	0.26

Abbreviations: CT—computed tomography, n—number, PE—pulmonary embolism.

**Table 2 jcm-11-03298-t002:** Comparison for mortality depending on PE status. Data are presented as number (median, percentage) followed by interquartile range, where applicable. *p*-values are presented after a comma.

	PE (−) N = 368; Death = 101 vs. Survivors = 267	PE (+) N = 88, Death = 19 vs. Survivors = 69
Age (years)	74 (65–82) vs. 67 (58–75), *p* < 0.001	73 (70–81) vs. 66 (55–77), *p* = 0.006
Male (n, %)	70 (70%) vs. 160 (60%), *p* = 0.09	14 (73%) vs. 50 (72%), *p* = 0.9
Length of follow-up for all (days)	6 (2–10) vs. 10 (5–15), *p* < 0.001	9 (5–12) vs. 12 (7–16), *p* = 0.06
CT lung involvement (%)	50 (40–75) vs. 40 (20–60), *p* < 0.001	30 (20–75) vs. 50 (30–60), *p* = 0.62
D-dimer (ng/mL)	2014 (906–4549) vs. 959 (553–1594), *p* < 0.001	7497 (2326–30,484) vs. 6443 (1921–17,726), *p* = 0.28
Troponin T (ng/mL)	0.056 (0.024–0.201) vs. 0.02 (0.009–0.059), *p* < 0.001	0.040 (0.026–0.240) vs. 0.024 (0.012–0.08), *p* = 0.03
Mortality (n;%)	101; 27.4	19; 21.5
Bova points	NA	2 (2–3) vs. 2 (1–2), *p* = 0.04

Abbreviations: CT—computed tomography, n—number, PE—pulmonary embolism.

**Table 3 jcm-11-03298-t003:** Independent predictors of mortality in PE (+) COVID-19 (+) patients and PE (−) COVID-19 (+) patients. Data are presented as odds ratio followed by 95% confidence interval and *p*-value.

Predictor	PE (+) Patients OR; *p*-Value	PE (−) Patients OR; *p*-Value
Age (years)	**1.06 (1.00–1.11); 0.04**	**1.07 (1.03–1.08); <0.001**
CT lung involvement (%)	0.99 (0.97–1.02); 0.51	**1.04 (1.02–1.05); <0.001**
D-dimer (ng/mL)	1.000 (1.000–1.000); 0.2	**1.002 (1.0001–1.003); 0.002**
Troponin T (ng/mL)	1.06 (0.26–4.30); 0.93	1.6 (0.77–3.6); 0.20

Abbreviations: CT—computed tomography, OR—odds ratio, PE—pulmonary embolism. Statistically significant results are presented in bold.

## Data Availability

Data available from the corresponding author upon written request.

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
