# Peer review of "Age-Adjusted D-Dimer Levels May Improve Diagnostic Assessment for Pulmonary Embolism in COVID-19 Patients"

_jcm, 2022, doi:10.3390/jcm11123298_

Round 1

Reviewer 1 Report

The manuscript by Machowski is timely topic and interesting. The methodology and statistical analysis is appropriate. I only suggest:

A minor revision for some typing errors.

To add the following reference: Zuin M, Rigatelli G, Zuliani G, Roncon L. Age-adjusted D-dimer cutoffs to guide anticoagulation in COVID-19. Lancet. 2021 Oct 9;398(10308):1303-1304. doi: 10.1016/S0140-6736(21)01859-6. PMID: 34627487; PMCID: PMC8497034. And discuss whether age-adjusted D-dimer may improve the PE treatment and/or anticoagulation in COVID-19 patients

Author Response

We thank the Reviewer for their comment. We are pleased to inform that we have adjusted the manuscript accordingly. All alterations are highlighted in the text.

1. We have added the suggested reference to the discussion section of the manuscript and have elaborated on the issue of anticoagulation treatment based on age-adjusted D-dimer levels (lines 242-246): (...)

Another noteworthy issue is the usefulness of guiding anticoagulation treatment with adjusted D-dimer levels [23], especially in the extended treatment (non-acute) phase. It has been reported that elevated D-dimer levels may be positively correlated with bleeding risk in non-COVID-19 PE patients [24], displaying potential in both prognosticating mortality and hemorrhagic complications during PE treatment. (...)

Reviewer 2 Report

Thank you for allowing me the opportunity to review this work. This is an excellent study with several important findings.

A few comments:

- Line 51 should be moved from the introduction into the methods.

- Please clarify if this is a cohort or case-control study and update the study accordingly (e.g. if case control then cannot use the word "incidence")

- How long were patients followed?

- Did any non-PE patient subsequently develop PE? How were these handled in the analysis statistically? Please indicate in the manuscript.

- It may be easier to present sex in table 1 as "Percent Female" instead of the counts currently shown.

- It is interesting that there is no difference in the troponin value between PE versus no PE. Does this mean that troponin may not be a reliable indicator of right heart strain in COVID positive PE patients? Were there differences in BNP? Please address in the manuscript.

- For the ROC curves, the arrows are in the margin of the paper (the arrows are not even on the figure). Please correct.

- Please provide information about when the d-dimer level was collected in relation to the diagnosis of PE.

- Please indicate the date ranges for the study.

Author Response

We thank the Reviewer for their comments. We are pleased to inform you that we have adjusted the manuscript accordingly. All alterations are shown in the text either using the track changes mode or highlighted in yellow.

Comment 1: Line 51 should be moved from the introduction into the methods.

Reply: Done, the sentence may now be found in lines 59-60.

Comment 2. Please clarify if this is a cohort or case-control study and update the study accordingly (e.g. if case control then cannot use the word "incidence")

Reply. Thank you for drawing attention to this important distinction. This is a retrospective cohort study (exposure: Sars-CoV-2, outcome: PE).

Comment 3. How long were patients followed?

Reply: The patients were followed-up for the length of hospitalization, median 10 days, IQR 5-14 days. This information is now highlighted in the text (line 141).

Comment 4: Did any non-PE patient subsequently develop PE? How were these handled in the analysis statistically? Please indicate in the manuscript.

Reply: The patients were followed-up only during hospitalization. None of the patients from the original non-PE group developed PE during in-hospital observation. This information has been added to the manuscript (line 142).

Comment 5:  It may be easier to present sex in table 1 as "Percent Female" instead of the counts currently shown.

Reply. Done.

Comment 6. It is interesting that there is no difference in the troponin value between PE versus no PE. Does this mean that troponin may not be a reliable indicator of right heart strain in COVID positive PE patients? Were there differences in BNP? Please address in the manuscript.

Reply: Thank you for commenting on this interesting, yet preliminary finding. It may be plausible that in this group of patients, the extent of SARS-CoV-2 infection (myocardial damage) were the main driver of elevated TnT.  However, taking into account certain limitations of this study, namely that consecutive patients, regardless of the risk of early PE mortality defined according to the ESC criteria were included we abstain from drawing any definitive conclusions based on the fore mentioned finding.  Unfortunately, we do not have data on NT-proBNP levels. We have supplemented the manuscript with this comment (lines 247-253).

Comment 7: For the ROC curves, the arrows are in the margin of the paper (the arrows are not even on the figure). Please correct.

Reply: Thank you for pointing out this editing error, which occurred unexpectedly when the original manuscript was copied into the template. The arrows are now positioned correctly.

Comment 8: Please provide information about when the d-dimer level was collected in relation to the diagnosis of PE.

Reply: D-dimer concentrations were quantitatively measured on the day of admission as part of a standard diagnostic protocol. In the case of a clinical suspicion of PE, a contrast-enhanced multi-slice computed tomography of pulmonary arteries (CTPA) was performed. The median delay between D-dimer measurement and PE diagnosis was 1 day.

Comment 9: Please indicate the date ranges for the study.

Reply: This study was conducted over eight months, from October 2020 to May 2021. This information is now highlighted in the text (line 57-58).